# Conditioning on "and nothing else": Simple Models of Missing Data between Naive Bayes and Logistic Regression

**David Poole** [1]   **Ali Mohammad Mehr** [1]   **Wan Shing Martin Wang** [1]

## Abstract

In situations where people report in a free-form way, we need to condition on the fact that someone *did not* report something. While we need to take into account that something was not reported, often there are too many statements that could be reported to consider each one; we only want to reason about those that were reported. In this paper we start with two simple, common models, namely Naive Bayes and logistic regression, which are equivalent models that are trained differently as to how missing data is handled. Naive Bayes is traditionally trained in a generative way, to make optimal predictions assuming only one value is observed (and making independence assumptions for the rest) and logistic regression is traditionally trained in a discriminative way, assuming no data is missing. It is generally assumed that these are qualitatively different, but in this paper we show there is a continuum between them. In particular, we show a model that is more general than both, but still simple, that can be trained to condition on missing data. In particular, it conditions on "and nothing else [was reported]" enabling us to avoid reasoning about the myriad of things that were not reported, but still take them into account.

## 1. Introduction

Bayesian conditioning is subtle in that one must condition on all available evidence. That fact that someone did not specify something is evidence that should be taken into account.

**Example 1:** Consider an application where someone volunteers information about themself, or a biographical knowl-edge graph about a person. In general:

$$P(has\_sibling) \neq P(has\_sibling \mid sibling\_was\_not\_mentioned)$$

because people often mention having a sibling if they have one, and don't mention a sibling if they don't have one. For example, if half the people have siblings, and siblings were mentioned in 90% of the cases where there were siblings and never mentioned when there were no siblings, $P(has\_sibling) = 0.5$ but $P(has\_sibling \mid sibling\_was\_not\_mentioned) = 5/55 \approx 0.09$.

Similarly, if patients typically specify a headache when they have one but it is not mentioned otherwise, not specifying a headache is information that needs to be conditioned on. Such cases are far from missing at random. We need to condition on the fact that something was not mentioned. In general, we need to be careful about the language we use to specify observations (Halpern, 2003).

There are typically too many things that were not mentioned, particularly in relational cases and cases where there are combinatorially many possible statements. In some sense we need a way to specify some formula is "all I know" (Levesque, 1990), or that some formula "and nothing else" was observed. This paper is about how to specify "and nothing else" for simple models.

One could build a sophisticated model of missing data (Little & Rubin, 1987; Marlin et al., 2011; Shpitser et al., 2015), but many applications are built with much simpler representations. Two common simple probabilistic representations, from which other representations are often built, are naive Bayes and logistic regression. For example, a common relational model, Markov logic networks (Domingos & Lowd, 2009) and its directed counterpart, relational logistic regression (Kazemi et al., 2014), often become equivalent to naive Bayes or logistic regression (Poole et al., 2012).

In this paper, we only consider Boolean variables, and no zero probabilities. We write Boolean variables in upper case, e.g., $X_i$, where true is represented as 1 and false as 0. Lower case variant, e.g., $x_i$, is the proposition $X_i = 1$, and $\neg x_i$ is the proposition $X_i = 0$. Extending this to multi-valued discrete variables (multinomials) is straightforward (using a softmax and indicator variables), but some of the results do not hold

[1]Department of Computer Science, University of British Columbia. Correspondence to: David Poole <poole@cs.ubc.ca>.

*Presented at the first Workshop on the Art of Learning with Missing Values (Artemiss) hosted by the 37th International Conference on Machine Learning (ICML).* Copyright 2020 by the author(s).

for continuous variables.

Suppose we want a representation for $P(Y \mid X_1 \ldots X_n)$. Logistic regression uses $n+1$ weights, with

$$P(y \mid X_1 \ldots X_n) = sigmoid(w_0 + w_1 X_1 + \cdots + w_n X_n)$$

where $X_i$ is a 0/1 variable that is 1 when $X_i$ is true and 0 when $X_i$ is observed to be false, and $y$ means $Y = true$.

Naive Bayes uses $P(Y)$, and $P(X_i \mid Y)$ for each $i$ (i.e., $2n+1$ parameters to represent the model). Note that Naive Bayes is a model of a distribution, and logistic regression is a model of a conditional distribution, but they can be compared when the $X_i$ are observed.

We show in Section 2 that when the $X_i$ are all observed, these are equivalent models in that they can be mapped to and from each other.

Naive Bayes and logistic regression are typically trained (Ng & Jordan, 2002) with different objective functions; naive Bayes is trained in a generative way, and logistic regression in a discriminative way. In particular, Naive Bayes is trained to optimize the case when only one of the $X_i$ is observed. That is, $P(Y)$ and $P(X_i \mid Y)$ for each $X_i$ are optimized independently, and so modularly, whereas logistic regression is trained for the case where all of the $X_i$ are observed. In this paper, we look at training essentially the same model for the case where some of the $X_i$ are observed. Both naive Bayes and logistic regression are able to be represented, but it can be trained to make predictions with missing data. In particular, we show that there is an interesting intermediate case between the generative and discriminative models.

This paper focuses on data that is Missing Not At Random (MNAR) and discrete. Rubin (1976) emphasizes the need to distinguish between Missing at Random (MAR) and MNAR data. For continuous data, Mohan et al. (2018) and Tang et al. (2003) apply regression to continuous missing data. The former focuses primarily on MNAR data, while the latter numerically computes the MLE, with constraints depending on the structure of missingness.

## 2. Equivalence of Models for Conditional Predictions

Here we give some basic results on probability and logistic regression. These results are/should be well known; "In both the discrete and continuous cases, it is well known that the discriminative analogue of naive Bayes is logistic regression" (Ng & Jordan, 2002).

First, $P(y \mid X_1 \ldots X_n) = sigmoid(\log odds(y \mid X_1 \ldots X_n))$

For a naive Bayes

$$P(y, X_1 \ldots X_n) = P(y) \prod_{i=1}^{n} P(X_i \mid y)$$

So

$$odds(y \mid X_1 \ldots X_n) = \frac{P(y, X_1 \ldots X_1)}{P(\neg y, X_1 \ldots X_1)}$$

$$= \frac{P(y)}{P(\neg y)} \prod_{i=1}^{n} \frac{P(X_i \mid y)}{P(X_i \mid \neg y)} \qquad (1)$$

$$\log odds(y \mid X_1 \ldots X_n) = \log \frac{P(y)}{P(\neg y)} + \sum_{i=1}^{n} \log \frac{P(X_i \mid y)}{P(X_i \mid \neg y)}$$

Note that in the log-odds sum above, there is one value in the sum for each $X_i$. From this, we can derive the bias $w_0$ and any $w_i$ for $i > 0$. Details are provided in the appendix.

These are all numbers that are specified as part of the naive Bayes model. Note that if the model does not obey the independence of the graphical model for naive Bayes, logistic regression can learn different parameters.

**Theorem 1.** *Any naive Bayes model for $P(Y, X_1 \ldots X_n)$ (with root $Y$) has an equivalent logistic regression model for the conditional distribution $P(Y \mid X_1 \ldots X_n)$.*

A constructive proof is in the appendix.

**Theorem 2.** *Any logistic regression model for $P(Y \mid X_1 \ldots X_n)$ has many equivalent naive Bayes model with the same conditional probability when the $X_i$ are observed.*

Given the values of $w_i$, for $i > 0$, there is one weight and two parameters to be assigned. One of them, say $P(x_i \mid \neg y)$ can be assigned arbitrarily, and the other is

$$P(x_i \mid y) = \frac{1}{1 + e^{-w_1 - \log odds(x_i \mid \neg y)}}$$

which is always in the range [0,1] because the exponential is always positive. Given $w_0$ and the values given or assigned above, $P(y)$ can be set to satisfy the above condition.

## 3. Missing Data

Naive Bayes allows arbitrarily many of the $X_i$ to be missing, and so provides a model of how to handle missing data. The graphical model of naive Bayes (Pearl, 1988) gives the independence that the $X_i$ are independent of each other given $Y$. But the naive Bayes trained traditionally also gives a model of missingness, which might not be appropriate even if the conditional probabilities are appropriate.

Naive Bayes has $2n+1$ parameters as opposed to the $n+1$ parameters for logistic regression. These extra parameters allow it to handle missing observations.

Consider what happens in naive Bayes when $x_i$ is observed. As shown in Equation (1), the contribution to odds – the number that is multiplied by the odds as compared to if $X_i$ was not mentioned in the model – is $\frac{P(x_i \mid y)}{P(x_i \mid \neg y)}$; call this value

$\alpha_i$. When $\neg x_i$ is observed, the contribution to the odds is $\frac{1-P(x_i|y))}{1-P(x_i|\neg y)}$; call this value $\beta_i$. Given an arbitrary $\alpha_i$ and $\beta_i$, we can solve for $P(x_i \mid y)$ and $P(x_i \mid \neg y)$, giving:

$$P(x_i \mid y) = \frac{\alpha_i(1-\beta_i)}{\alpha_i - \beta_i} \quad , \quad P(x_i \mid \neg y) = \frac{1-\beta_i}{\alpha_i - \beta_i}$$

Because all of the probabilities are on the range $[0,1]$, it is straightforward to show that

**Theorem 3.** *One of the following must hold for each $\alpha_i$ and $\beta_i$: (1) $\alpha_i > 1$ and $\beta_i < 1$ (2) $\alpha_i = 1$ and $\beta_i = 1$ (3) $\alpha_i < 1$ and $\beta_i > 1$*

We can now build a logistic regression model that corresponds to the naive Bayes model, including the way to handle missing data. We call this a logistic regression($\pm$) model. Each Boolean random variable $X_i$ is represented by two weights: $w_i^+$ used when $X_i$ is observed to be true, and $w_i^-$ used when $X_i$ is observed to be false.

The model is defined by having two indicator functions for each $X_i$: $X_i^+$ has value 1 when $X_i$ is observed to true and value 0 otherwise, and $X_i^-$ has value 1 when $X_i$ is observed to false and value 0 otherwise. Note that if there are no observations for $X_i$, then $X_i^+$ and $X_i^-$ are both zero.

Then we have

$$P(y \mid X_1 \ldots X_n \text{ and nothing else})$$
$$= sigmoid(w_0 + \sum_{i=1}^{n} w_i^+ X_i^+ + w_i^- X_i^-)$$

This is a logistic regression model that also covers naive Bayes. In particular, to represent a naive Bayes model, using the notation above, $w_i^+ = \log \alpha_i$ and $w_i^+ = \log \beta_i$.

In naive Bayes, trained in a generative way, the probability that is predicted when nothing is observed is $P(y)$, but here it is $sigmoid(w_0) = P(y \mid \text{nothing was observed})$.

We can now learn this model in a generative way that takes missing information into account. Theorem 3 has an interesting consequence. $w_i^+$ and $w_i^-$ are not independent parameters, but have to be of opposite signs. The way to think about them is that a prediction that has $X_i^+ = X_i^- = 0$ has not observed anything about $X_i$. If observing $X_i = true$ has a positive effect, then observing it is false must have a negative effect, and vice versa. If observing $X_i = true$ has no effect (so $w_i^+ = 0$) then observing $X_i = false$ must also have no effect[1].

**Example 2**: Consider how to represent a model where we want to model how *HappyAlone* depends on *HaveSibling*

and *Anxiety*. Suppose the ground truth of $P(HappyAlone \mid HaveSibling, Anxiety)$ can be represented as either naive Bayes (Figure 1 (a)) or logistic regression (Figure 1 (b)).

Suppose we have, for the Naive Bayes model, $P(happyAlone) = 0.6$, $P(haveSibling \mid happyAlone) = 0.3$ and $P(haveSibling \mid \neg happyAlone) = 0.8$, $P(anxious \mid happyAlone) = 0.2$, $P(anxious \mid \neg happyAlone) = 0.5$. Then $P(haveSibling) = 0.5$, and $P(anxious) = 0.3$.

The logistic regression model with the same $P(HappyAlone \mid HaveSibling, Anxiety)$ is $sigmoid(2.13 + -2.23 * HaveSibling - 1.38 * anxious)$. The logistic regression model of Figure 1 (b) with $P(haveSibling) = 0.5$, and $P(anxious) = 0.3$, has the same marginals *HaveSibling* and *Anxious*, and the same conditional distribution as the Naive Bayes model, but the distributions are different. In particular, *HaveSibling* and *Anxious* are independent in the model of Figure 1 (b), but are not independent in the naive Bayes model.

Suppose also that *HaveSibling* and *Anxiety* are sometimes missing, but are not missing at random. In particular *HaveSibling* is missing as in Example 1, and *Anxiety* is missing with people rarely reporting that they don't have anxiety. Given *Anxiety* is true, it is reported 80% of the time, and Given *Anxiety* is false, it is reported 10% of the time. Assume the people are honest; that is they never report a value that isn't accurate.

Using the method of (Mohan et al., 2013; Marlin et al., 2011), for each variable $X$ there is a Boolean variable which we write $RepX$, that is true when $X$ was reported, and a variable $X^*$ which has the domain of $X$ together with an extra value meaning there was no report. For Boolean $X$, the range of $X^*$ is $\{true, false, no\_report\}$. The parents of $X^*$ are $X$ and $RepX$, and $P(X^* \mid X, RepX)$ is a deterministic function: the value of $X^*$ is the value of $X$ if $RepX = true$; otherwise it is $no\_report$. For simplicity, assume that what was reported for $X$ only depends on the value for $X$; that is $X$ is the only parent of $RepX$. Augmenting the previous two models with this is depicted in the belief networks of Figure 1 (c) and (d). For the experiments below we assume that $RepX$ is marginalized out.

A $LR\pm$ model corresponding to this example has five parameters, as follows:

$$P(happyAlone) = sigmoid(w_0 + w_1^+ haveSibling^+$$
$$+ w_1^- haveSibling^- + w_2^+ anxiety^+ + w_2^- anxiety^-)$$

Note that $haveSibling^+$ corresponds to the characteristic function for $HaveSibling^* = true$ and $haveSibling^-$ corresponds to $HaveSibling^* = False$. When *HaveSibling* is not observed, both are zero. We evaluate the effectiveness of this in the results section:

---

[1]Recall that we are assuming no zero probabilities. If we allow zero probabilities, the closed world assumption (Reiter, 1978), where a value is false if not specified, could be used in which case one of the $w$s could be zero. However, the closed world assumption provides a very blunt instrument for handling missing values.

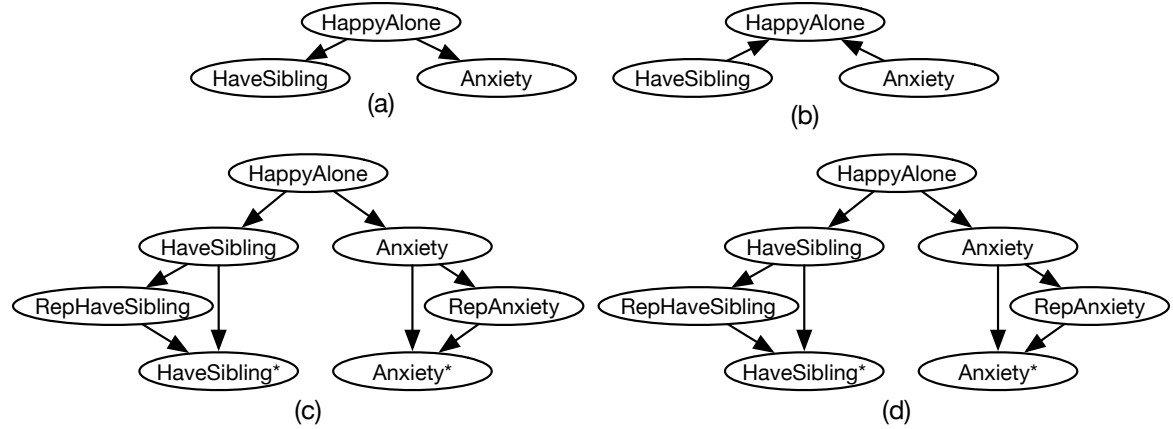

*Figure 1.* Belief networks from Example 2

## 4. Evaluation

For evaluation, we performed three main sets of experiments, generating 600 synthetic datasets in total - each with 20000 training samples and 5000 test samples - based on ground truth diagrams c) and d) in Figure 1. We evaluate on:

- 200 datasets generated for when the ground truth is the graphical model of (c) - we will call them C datasets,

- 200 datasets for when the ground truth is the graphical model of (d) where P(HappyAlone | HaveSibing, Anxiety) is a logistic regression (with 3 free parameters) - we will call them DLR datasets,

- 200 datasets for when the ground truth is the graphical model of (d) where P(HappyAlone | HaveSibing, Anxiety) is a table (with 4 free parameters) - D4 datasets.

We trained LR±, Naive Bayes, model (c) and model (d) where P(HappyAlone | HaveSibing, Anxiety) is a table with 4 free parameters (Full-(d) in table below). Models (c) and Full-(d) were trained using a maximum of 15 steps of expectation maximization (EM). In the models of (c) and (d) we assume that *RepHaveSibling* and *RepAnxiety* are marginalized out. In the following, for each set of datasets, we compare the logloss of every model compared to the logloss of ground truth model; the numbers are the average percentage increase in logloss compared to the ground truth model (lower is better):

| Dataset | Naive Bayes | Model (c) | Full (d) | LR± |
|---------|-------------|-----------|----------|------|
| C | 1.9% | 1.0% | 1.9% | 0.1% |
| DLR | 0.9% | 0.9% | 1.0% | 0.1% |
| D4 | 2.7% | 2.3% | 1.8% | 1.6% |

Note that LR± is always better on average. Despite the relative simplicity of our model, we can obtain log-loss errors

better than Naive Bayes and the other more complex models that require EM to train. EM does not always converge to a global optima, which explains why the trained model (c) did not consistently get an optimal logloss on C datasets and trained full-(d) model did not consistently get an optimal logloss on D4 datasets. They should do better with multiple restarts on EM, which we did not do. We also observed that in all of the datasets, the comparison made by Ng & Jordan (2002) holds even in missing data setting: LR± always has less logloss compared to Naive Bayes in our datasets, which have large number of samples. Box plots showing the distributions of the above improvements and a link to the code are in the appendix.

## 5. Conclusions

This paper started off from building applications where we wanted a simple model, but where most of the potential observations were missing and they were not missing at random. They were missing because the person providing the information did not think they were worth reporting (e.g. they they did not have a headache) or that they were not something they thought of (e.g., that they do not have a sibling or that their finger is still connected to their hand). Logistic regression did not seem applicable, and naive Bayes gives the wrong answer (for the reasons explained above). The obvious solution was to include weights for both positive and the negative for each Boolean condition, and the bias was the probability when nothing was observed. We were surprised that this is not a common and well-understood model.

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

# Appendices

## A. Equivalent Logistic Regression and Naive Bayes models

The derivation from section 2, of the equivalent Logistic Regression to the Naive Bayes model is provided here: (Ng & Jordan, 2002)

As long as all probabilities are positive (no zero probabilities)

$$P(h \mid e) = \frac{P(h \wedge e)}{P(e)}$$

$$= \frac{P(h \wedge e)}{P(h \wedge e) + P(\neg h \wedge e)}$$

$$= \frac{1}{1 + P(\neg h \wedge e)/P(h \wedge e)}$$

$$= \frac{1}{1 + e^{-\log \frac{P(h \wedge e)}{P(\neg h \wedge e)}}}$$

$$= sigmoid(\log odds(h \mid e))$$

Where $sigmoid(x) = 1/(1 + e^{-x})$ and

$$odds(h \mid e) =_{def} \frac{P(h \wedge e)}{P(\neg h \wedge e)}$$

$$= \frac{P(h \mid e)}{P(\neg h \mid e)}$$

$$= \frac{P(h \mid e)}{1 - P(h \mid e)}$$

We will now show that for a naive Bayes, odds is a product and so the log-odds is a sum of one value for each $x_i$.

## B. Derivation of bias and weights

The bias and weights from section 2 are derived here.

The bias $w_0$ is the number used when all of the $x_i$ are false:

$$w_0 = \log \frac{P(y)}{P(\neg y)} + \sum_{i=1}^{n} \log \frac{P(\neg x_i \mid y)}{P(\neg x_i \mid \neg y)}$$

$$= \log \frac{P(y)}{1 - P(y)} + \sum_{i=1}^{n} \log \frac{1 - P(x_i \mid y)}{1 - P(x_i \mid \neg y)}$$

For $i > 0$, $w_i$ is the value that is added when $x_i$ is true.

$$w_i = \log \frac{P(x_i \mid y)}{P(x_i \mid \neg y)} - \log \frac{P(\neg x_i \mid y)}{P(\neg x_i \mid \neg y)}$$

$$= \log \frac{P(x_i \mid y)}{1 - P(x_i \mid y)} \frac{1 - P(x_i \mid \neg y)}{P(x_i \mid \neg y)}$$

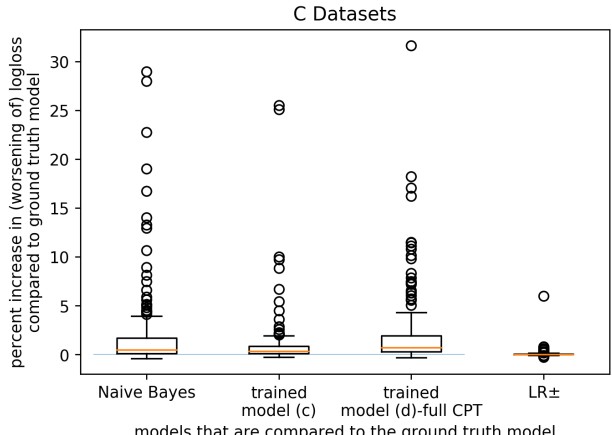

*Figure 2.* Box plots comparing logloss for every model with the ground logloss for truth model in C datasets

## C. Extended Experiment results

In Figures 2, 3 and 4, for each set of datasets, box plots are shown comparing the logloss for every model with the logloss for the ground truth model for those datasets.

For the synthetic datasets used in this analysis, data was generated by sampling from categorical distributions. The parameters of these distributions are drawn from uniform probability distributions, and the sampling procedure depends on the relevant graph structure: Figure 1c) for dataset C, and Figure 1d) for the DLR and D4 datasets.

For reference, example 2 contains one set of categorical parameters corresponding to figure 1c).

Specifically, as described in section 4, we had 600 synthetic datasets (200 for each of C, DLR, D4), each with 20000 and 5000 training and test points respectively.

The expectation maximization for training models (c) and (d) was implemented using PyMC3 by Salvatier J (2016). Naive Bayes and LR± was implemented using scikit-learn by Pedregosa et al. (2011). Our code is available on GitHub at https://github.com/alimm1995/logistic-regression-pm.

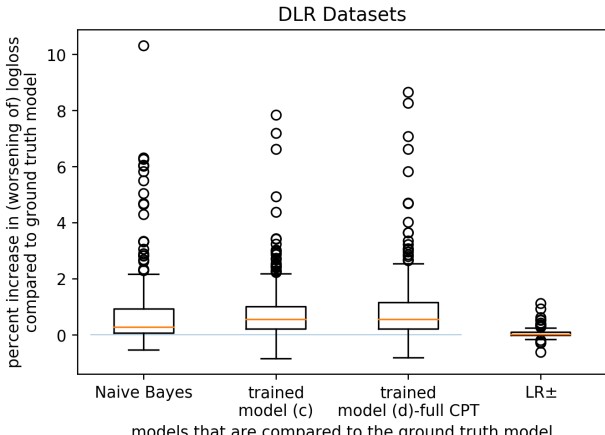

*Figure 3.* Box plots comparing logloss for every model with the logloss for ground truth model in DLR datasets

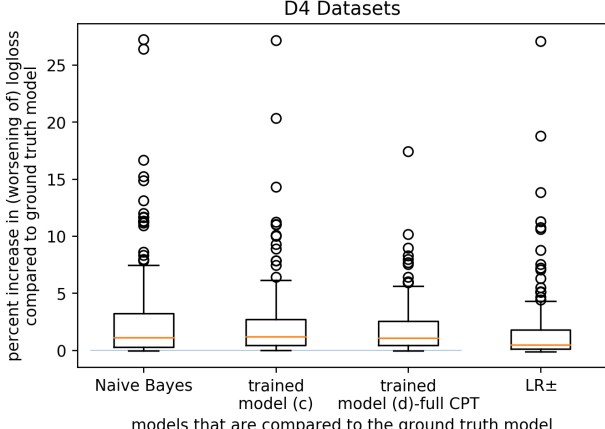

*Figure 4.* Box plots comparing logloss for every model with the logloss for ground truth model in D4 datasets