# OpenReview forum: "Conditioning on "and nothing else": Simple Models of Missing Data between Naive Bayes and Logistic Regression"
_ICML.cc/2020/Workshop/Artemiss — ICML Artemiss 2020_

### Official Review · AnonReviewer2 · 2020-06-22
**The authors present a method to deal missing not at random data, for Boolean variables (or multinomial), without explicitly modeling the missingness process for the data.**

**Rating:** 7
**Confidence:** 4

**Review:**

The authors present a method to deal missing not at random data, for Boolean variables (or multinomial), without explicitly modeling the missingness process for the data. They use a logistic regression, which corresponds to the naive Bayes model by including weights for both positive and the negative for each Boolean condition to handle missing data.

I think this paper is of interest for the workshop.

It would have been interesting to give the percentage of missing data in the simulation.

The literature in MNAR (Missing Not At Random) data can be expanded. To introduce the missingness processes for data and the importance of being taking into account in the MNAR setting, the work of Rubin 1996 Inference and missing data could be cited.
In addition, for continuous variables, some existing works also handle MNAR data in simple models (multivariate regression models, linear models) whithout explictly modeling the missingness process for the data (as Mohan, Estimation with incomplete data: The linear case or Tang Analysis of multivariate missing data with nonignorable nonresponse).

Besides, there is a compilation error in Appendix C for " in figures C,C and C..."

---

### Official Review · AnonReviewer1 · 2020-06-23
**simple and comparable**

**Confidence:** 4
**Rating:** 6

**Review:**

This work shows an interesting intermediate case between the generative and discriminative models. Simple model structure and it is able to obtain comparable log-loss errors to Naive Bayes and the other more complex models.

Maybe can elaborate more details about how to generate synthetic datasets and add more expriments using real data set.

Pay a little attention to the Appendices C, I think there is a mistake.

---

### Decision · Program_Chairs · 2020-07-02

**Decision:**

Accept

**Comment:**

We are very happy to inform you that your paper has been accepted for the Artemiss workshop. We will contact you soon to inform you about the details concerning the format of your presentation at the workshop, and the camera-ready version deadline. Please take into account the referee's comments to write the camera-ready version.